# Sorption-Assisted Ultrafiltration Hybrid Method for Treatment of the Radioactive Aqueous Solutions

Leon Fuks, Agnieszka Miśkiewicz * and Grażyna Zakrzewska-Kołtuniewicz

Centre for Radiochemistry and Nuclear Chemistry, Institute of Nuclear Chemistry and Technology, Dorodna 16, 03-195 Warszawa, Poland
* Correspondence: a.miskiewicz@ichtj.waw.pl

**Abstract:** The paper presents results of studies on the possibility of using the ultrafiltration method supported by sorption on low-cost, easily accessible aluminosilicates to purify water contaminated with radionuclides. An aqueous solution contaminated with radionuclides in the form of cations at different oxidation states—Cs(I)-137, Co(II)-60 and Am(III)-241—as well as pertechnetate anions—$TcO_4^-$-99m—was treated by the proposed hybrid method. In the presented work, the influence of the important process parameters (i.e., pH, sorbent dosage, temperature and feed flow rate) on the removal efficiency of radionuclides was studied. The obtained results showed that hazardous impurities, both in the form of cations and anions, may be effectively removed from water by the application of sorption-assisted UF (SAUF) using the clay-salt slimes as a sorbent. As a final stage of the work, we treated the simulated liquid radioactive waste using the SAUF method, also showing satisfactory results in its purification efficiency.

**Keywords:** water contamination; radioactive waste; decontamination; sorption-assisted ultrafiltration; SAUF; low-cost adsorbent; clay–salt slimes

## 1. Introduction

In the last few decades, there have been two nuclear accidents, classified as "Major Accidents" in the International Nuclear and Radiological Event Scale (INES) [1]. Namely, there were the Chernobyl disaster of 1986 and the Fukushima Daiichi nuclear accident in 2011. In both cases, several harmful radionuclides were released into an aqueous system and, then, contaminated the soil [2]. Specific radioactivity concentrations of the main radionuclides found in Pripyat River (the river that flows through the exclusion zone established around the site of the Chernobyl nuclear disaster) after the Chernobyl accident [3] exceeded their maximum acceptable concentrations in drinking water (according to the World Health Organization, WHO [4]), even by approximately 450 times. Namely, the radioactivity concentration of I-131 was detected to be 4440 Bq·L$^{-1}$ vs. 10 Bq·L$^{-1}$, Cs-137–1590 Bq·L$^{-1}$ vs. 10 Bq·L$^{-1}$, Sr-90–30 Bq·L$^{-1}$ vs. 10 Bq·L$^{-1}$, Zr-95–1550 Bq·L$^{-1}$ vs. 100 Bq·L$^{-1}$ and Pu-241–33 Bq·L$^{-1}$ vs. 10 Bq·L$^{-1}$ [3,4]. Thus, obtaining the consumption water required treatment by the effective purification methods.

A certain amount of the radioactive liquid waste is also generated during nuclear fuel production and its use in nuclear reactors, as well as the production and use of radioisotopes in the research, nuclear medicine, industry and agriculture. To ensure protection of the environment and humans from the hazard arising from this waste, the existing methods of the waste treatment should be continuously improved.

As the main conventional methods of treating the contaminated water, one should mention coagulation, sedimentation and sorption, all followed by filtration [5]. These procedures consist of dosing special chemicals (coagulant, hydrolyzing agent, sorbent, respectively) into water and allowing them to form species that are easy to separate. However, in the recent past, ultrafiltration (UF) has been recognized as the most important

technology in water treatment. The main advantages of ultrafiltration are the little space required for the installation of the equipment, easy control of the quality of the treated water independent of the feed quality, and simple automation. However, the removal of the contaminants by the UF process takes place because of the size exclusion by the membrane, with pores that are of sizes ranging from 0.1–0.01 μm. This means that either mono- or multivalent ions can pass through the membrane and cannot be completely separated from water. In turn, high molecular compounds do not penetrate through the membrane and are deposited the surface.

The possibility of separating small ions by binding them to water-soluble polymers (e.g., chitosan, sodium alginate) has been proposed and studied [6,7]. The results appeared satisfactory and presented the following advantages over the conventional methods: (1) the process does not require the dosage of additional components into water, (2) it allows for treating larger volumes of water than nanofiltration (NF), liquid–liquid extraction, sorption or ion exchange, and (3) both the polymer and membrane may be regenerated and reused. In the industrial scale, ultrafiltration enhanced by complexation with water-soluble polymers was used at Los Alamos National Laboratory for the treatment of liquid radioactive waste arising from the decommissioning and decontamination of nuclear facilities [8].

Water-insoluble polymers (ion exchangers, sorbents) have also been intensively tested, and, as a result, the hazardous metal cations were rejected from the contaminated water [9].

The data available in the literature indicate that one of the first attempts to purify radioactively contaminated water by ultrafiltration, preceded by using a solid ion exchanger, was presented in 1991 by Hooper [10]. In a small (laboratory) scale ultrafiltration unit, the authors used sodium nickel hexacyanoferrate (II) and manganese dioxide as a solid ion exchange/sorbent followed by ultrafiltration in a small (laboratory)-scale unit to treat the laundry effluent contaminated with plutonium and americium. Soon after, similar work was carried out in Poland. In 1994, Chmielewski et al. reported results on the treatment of radioactive aqueous solutions using ultrafiltration preceded by mixing the solution with copper or nickel hexacyanoferrates [11] Later, such work was undertaken at the INCT and resulted in the papers of Zakrzewska et al. [12–14].

The application of this method at an industrial scale has also been tested, giving positive results. One of the first installations for the removal of radioactive metals from aqueous solutions in hybrid SAUF systems was performed at the Cadarache Nuclear Research Centre (France), where ultrafiltration was combined with sorption on the activated carbon and nickel hexacyanoferrate [15]. Inorganic sorbents were also evaluated at Harwell, where ultrafiltration was used to treat wastewater from the PWR reactor [16–18].

Therefore, finding easily available, cheap and ionizing radiation-resistant sorbents ready to use in the SAUF process becomes an important problem [19]. The last of these requirements is related to the possibility of applying the proposed method for the treatment of liquid radioactive waste. This waste is usually an aqueous solution with radionuclides that emit ionizing radiation, sometimes of high activity. Currently presented work is included in the list of such examinations. The aluminosilicate material that was chosen as a potential sorbent is the powdered clay–salt slime (CSS). It is an industrial waste generated at the JSC "Belaruskali" (Soligorsk, Belarus), and it originated from the production of artificial fertilizers. The studies conducted already by our group on the radionuclide batch sorption using the CSS allow for suggesting conditions for carrying out the process so that it will be possible to recover radionuclides important in other fields (e.g., in nuclear medicine) [20]. It has also been shown that the sorbent is stable enough under an action of the ionizing radiation [21].

An important issue in the design of the SAUF process is the choice of the type of membrane to be used in the ultrafiltration process. From the many commercially available polymer membranes, based on our previous experience, we chose polyethersulfone ultrafiltration membranes. Their main advantages are resistance to the elevated temperature and $Cl_2$, as well as to the pH of the purified solution, and its ease of production. The PES material was produced in an advanced fiber-spinning process that allows for producing large and uniform sheets with densely distributed pores, the sizes of which result in an improvement in the selectivity of the separation of different species. Moreover, the investigations of the impact of the ionizing radiation on the performance of polyethersulfone ultrafiltration membranes revealed that there were no significant changes to the membrane surfaces within the tested dose range [22]. The above indicates the possibility of using the PES membranes for the filtration of radioactively contaminated water, as well as liquid low-level radioactive waste.

There are many radioactive metals that might contaminate water [23], and four of them were examined in the presented study. Those were cations: caesium-137 (as a representative for the monovalent cations), divalent cobalt-60 (which represents the metallic corrosion products), as well as the trivalent americium-241 (representing the nuclear fission products). Metallic oxoanions were represented by technetium-99m (chemical equivalent for technetium-99), which exists in the form of pertechnetates $TcO_4^-$. All of these radionuclides emit gamma radiation, and, therefore, their concentrations in water are relatively easy to determine.

## 2. Experimental Details

### 2.1. Chemicals and Sorbent

The sorbent (clay–salt slimes, CSS) was received in a form ready for experiments as a courtesy from Prof. Maskalchuk (Belarusian State Technological University Minsk, Belarus). The crude material, an industrial waste from the Joint Stock Company JSC "Belaruskali" (Soligorsk, Belarus), was taken from the depot No. 1, washed several times with the distilled water to remove the soluble salts (mainly NaCl and KCl) and dried at 50 °C to constant mass (for approximately 6 h). The sieve analysis showed that the sorbent was a fine grade material and that the main fraction had a diameter of 0.2 mm. Approximately 98% of the particles were greater than 0.05 mm [21].

All other chemicals (Sigma Aldrich/Fluka, Katowice, Poland), puriss. p.a., were used as delivered. Water used in the experiments was deionized.

Carrier-free radionuclides of caesium-137 ($t_{1/2}$ = 30.07 y; $E_\gamma$ = 661.7 keV), cobalt-60 ($t_{1/2}$ = 5.3 y; $E_\gamma$ = 1173.2 and 1332.5 keV) and americium-241 ($t_{1/2}$ = 432.2 y; $E_\gamma$ = 59.5 keV) were provided by the POLATOM (Otwock-Swierk, Poland) as certificated standard aqueous solutions. Working solutions of the desired concentrations of the radionuclides were prepared by gravimetric dilution of these standards with the nitric acid (pH 3) and checked for the radiochemical purity by gamma ray spectrometry.

Technetium-99m radionuclide ($t_{1/2}$ = 6.01 h; $E_\gamma$ = 140.5 keV), used instead of Tc-99, was eluted from a Mo-99/Tc-99m commercially available medical generator (GE Healthcare, supplied by Biker, Warsaw, Poland) as a 0.9% NaCl solution of a specific activity of approximately 100 MBq $cm^{-3}$. This radionuclide was used because of its favorable radiometric properties with respect to those of technetium-99. Radiochemical purity of the eluate was also checked by gamma ray spectrometry. Representative spectra are shown in Figure 1.

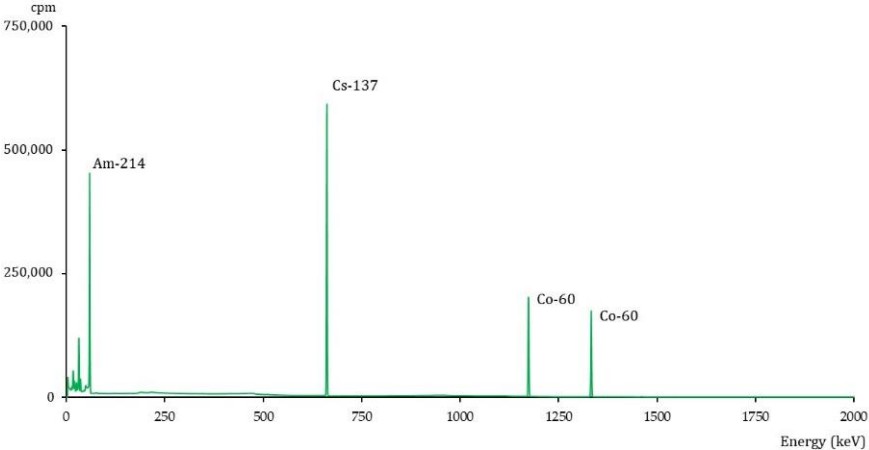

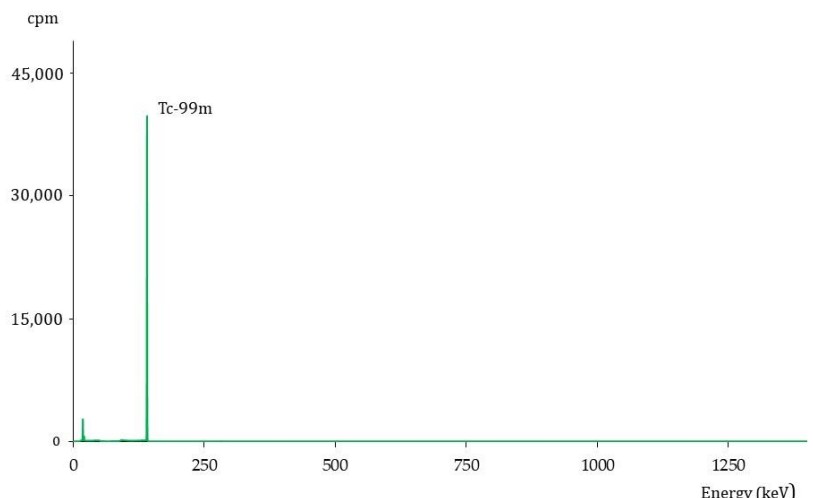

**Figure 1.** Gamma ray spectrum of the working solution containing Am-241, Cs-137 and Co-60 (**upper**) or Tc-99m (**bottom**).

### 2.2. Laboratory Ultrafiltration Kit

The ultrafiltration chamber used was a magnetically stirred membrane cell AMICON 8400 pressure-driven filtration compartment coupled with magnetic stirring that offers a broad range of process volumes (up to 400 mL). The chamber was equipped with a polyethersulfone ultrafiltration membrane ($C_6H_4$-4-$C(CH_3)_2C_6H_4$-4-$OC_6H_4$-4-$SO_2C_6H_4$-4-$O]_n$; CAS Number: 25135-51-7) with a molecular weight cut-off of 10 kDa, diameter of 7.6 cm and surface in contact with liquid of 41.8·$cm^2$. Both the ultrafiltration chamber and the membranes were delivered by Merck Millipore (Merck Millipore, Poznań, Poland).

Compressed nitrogen was applied as a pressure source to assure constant liquid flow through the membrane. The water flux ($J_w$) was determined experimentally prior to the UF/sorption process under transmembrane pressure of 2 bar. The measured $J_w$ values were in the range of 0.0164–0.0183 $m^3$ $m^{-2}$ $s^{-1}$).

Schematic presentation of the laboratory-scale installation is presented in Figure 2.

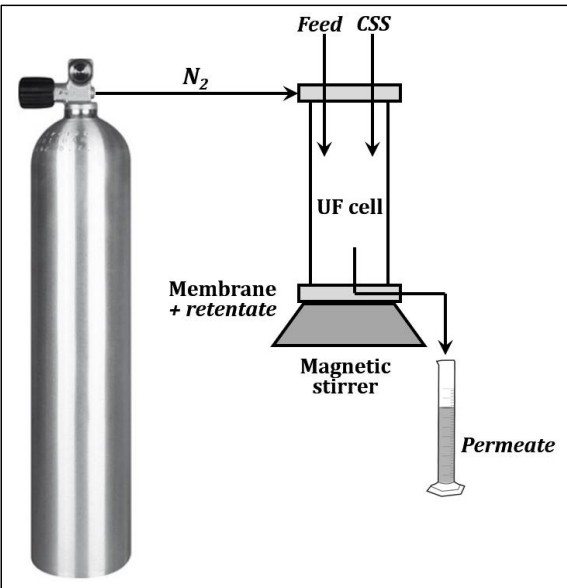

**Figure 2.** Schematic presentation of the laboratory-scale SAUF installation.

*2.3. Studies on the Water Purification*

The required amount of sorbent was added to the ultrafiltration cell containing radioactively contaminated solution and was magnetically stirred for 10 min. Then, an ultrafiltration process was forced by a stream of nitrogen. The permeate was fractionated into 10 mL portions and gravimetrically confectioned into 1 mL portions.

The specific radioactivity of any radionuclide present in the mixture was determined by applying a well-type counter with an automatic sample changer. The reliability of the measurements was periodically checked by measuring the adequate radionuclide standard solutions. The radiometric measurements of each sample were repeated three times and the results were presented as mean ± standard deviation.

Influence of the important process parameters, i.e., pH, CSS dose and phase contact time applied prior to ultrafiltration, on the removal efficiency of Cs-137, Co-60 and Am-241 radionuclides, was studied. A separate part of the work was to check the possibility of purification of aqueous solutions from the Tc-99m ions.

Efficiency of the removal of radionuclides ($E_{radionuclide}$) was determined by calculating the ratio, which takes into account concentration of the radioactive nuclide in the permeate (i.e., in the solution flowing out of the filtration cell) and its concentration in the feed solution:

$$E_{radionuclide} = \frac{A_f - A_p}{A_f} \cdot 100\% = \left(1 - \frac{A_p}{A_f}\right) \cdot 100\%$$

where $A_f$ and $A_p$ are the specific radioactivity concentrations of the feed and permeate, respectively, [Bq·mL$^{-1}$].

*2.4. Methods Used in the Work and the Relevant Instrumentation*

Methods used for examining the sorbent-assisted UF and the relevant devices used are shown in Table 1.

**Table 1.** Methods for examining the sorbent-assisted UF and the devices used in the present work.

| Description of the Experiment | Methods | Devices |
|---|---|---|
| Production of the saline aqueous solutions containing Tc-99m | Elution from the isotope generator; typical medical procedure | Mo-99/Tc-99m medical generator; GE Healthcare, (supplied by Biker, Warsaw, Poland) |
| Ultrafiltration (UF) | Sorbent-assisted ultrafiltration (UF) in the laboratory scale | The polyethersulfone (PES) ultrafiltration membrane (Merck Millipore) The magnetically stirred membrane ultrafiltration cell AMICON 8400 (Merck Millipore) |
| Radiometric analyses of water | Well-type radiometric counting of gamma radiation | Perkin Elmer 2480 Wizard2© Automatic Gamma Counter (Waltham, MA, USA) |
| Checking purity of the radionuclides | Gamma radiation spectrometry | Gamma ray spectrometer with HPGe semiconductor detector (ORTEC, Canberra, Australia) with efficiency from 30 to 45% |
| Checking chemical composition of aqueous solutions | Ion chromatography | Ion chromatograph DIONEX ICS-5000 DC (DIONEX, Sunnyvale, CA, USA) |

## 3. Results and Discussion

### 3.1. Purification of the Contaminated Water

#### 3.1.1. Removal of Cationic Metal Species: Cs(I), Co(II) and Am(III)

To check whether the sorption-assisted ultrafiltration (SAUF) may be successfully used as an efficient method for the purification of water contaminated with radionuclides, as well as low- and intermediate radioactive liquid wastes, in the presented work, we used the laboratory prepared aqueous solution containing monovalent- (Cs-137), divalent- (Co-60) and trivalent (Am-241) cations as the representatives of the metallic radionuclides that may contaminate water. A case result of water purification presenting the $E_{radionuclide}$ values along the UF permeate volume for a pH of 2.2 is shown in Figure 3 (left). In the experiment, the dosage of the CSS sorbent was 2.5 g·L$^{-1}$ and the flow rate of the feed water was kept within the range of 0.8–1.4 mL·min$^{-1}$ by a nitrogen stream pressure of 2 bar. The total amount of the collected permeate (viz. 160 mL), being equal to the volume of the mixing chamber, was fractionated in portions of 10 mL each. Figure 3 (right), in turn, shows the pattern of $E_{radionuclide}$ along the series of the purified solutions of different acidity. As one can observe, the pH value of the purified water changing in the range of 1–12 does not determine the removal of either mono- or multivalent cations. In detail, the $E_{radionuclide}$ values are 97.5 ± 1.4, 95.8 ± 1.3 and 97.9 ± 0.8 for Cs-137, Co-60 and Am-241, respectively.

To determine the optimum dosage of the sorbent, i.e., to find the quantity that guarantees the effective removal of the radionuclides from the contaminated water, several amounts of the CSS were examined. They were kept in the range from 0.5 to 3.0 g·L$^{-1}$, whereas the other parameters remained constant. The obtained results are presented in Figure 4. It can be seen that, when increasing the adsorbent dose up to 2.5 g·L$^{-1}$, for any metal, the removal efficiency approaches 100%. Therefore, it can be proposed that the sorbent dosage of 2.5 g·L$^{-1}$ is the optimum, and that we used it in further experiments.

It is well known that the water viscosity decreases with an increasing temperature, so the water flux passing the membrane increases with an increasing temperature [24,25]. While designing a technological process, the question arises over whether the efficiency of the water purification system may be enhanced by increasing the temperature of the process. Therefore, we performed the SAUF experiments at three rational temperatures: 15, 25 and 45 °C. The results are shown in Figure 5 (left). As can be seen, $E_{radionuclide}$ does not remarkably depend on the process temperature for both the acidic and basic solutions. Therefore, it can be suggested that running the process at an ambient temperature is equally reasonable from the point of view of the water purification effect and from its operating costs.

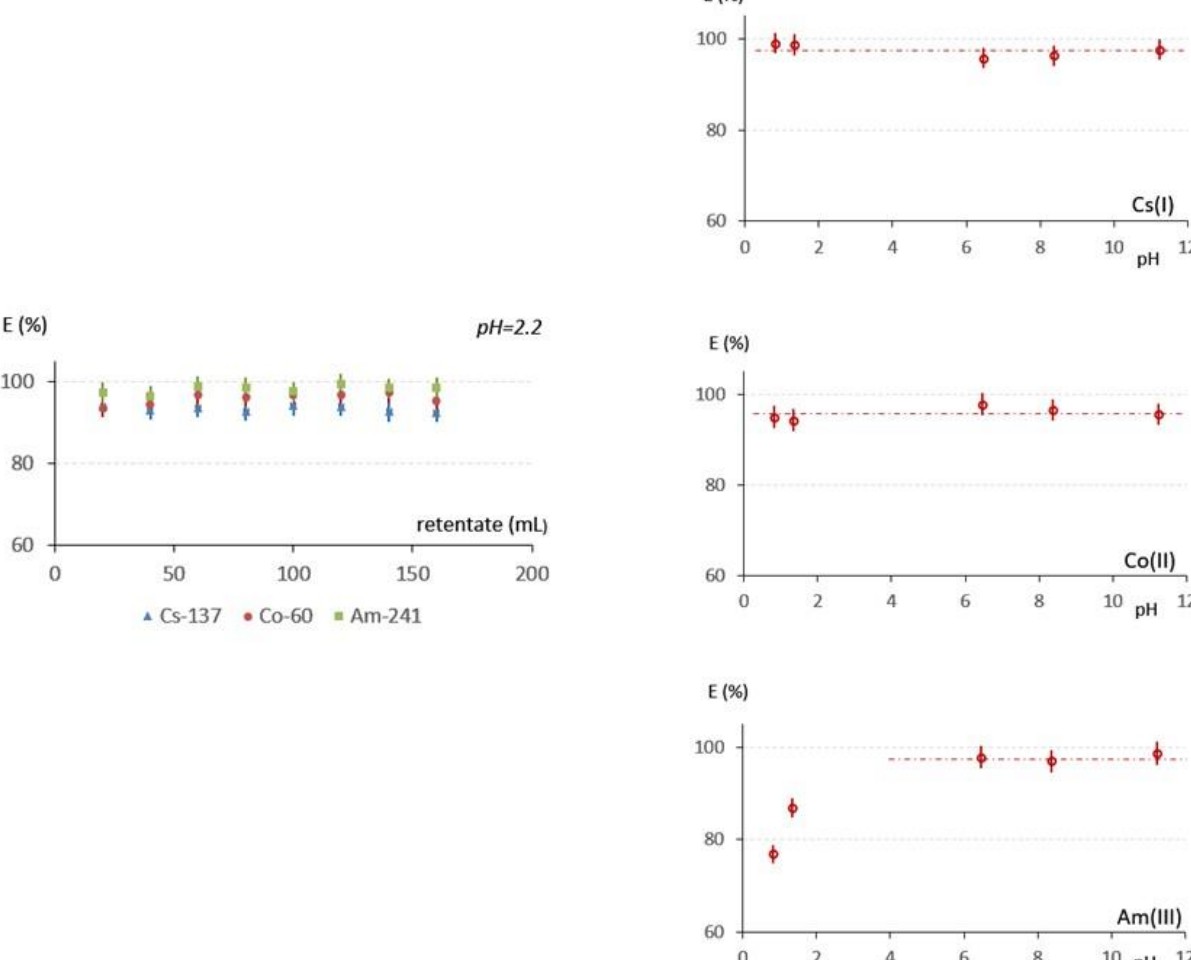

**Figure 3.** Removal of Cs(I), Co(II) and Am(III) radionuclides from simulated wastewater; an example run for pH = 2.2 (**left**) and the pH dependence (**right**).

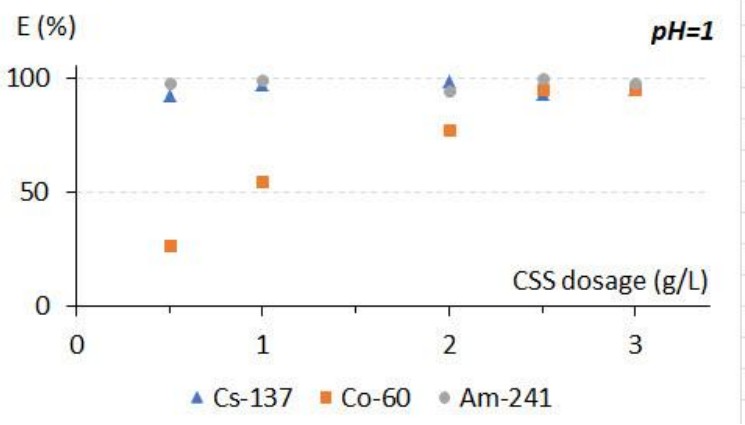

**Figure 4.** Dependence of the purification efficiency, E (%), of three radioactive metal cations on the CSS dose (g·L$^{-1}$).

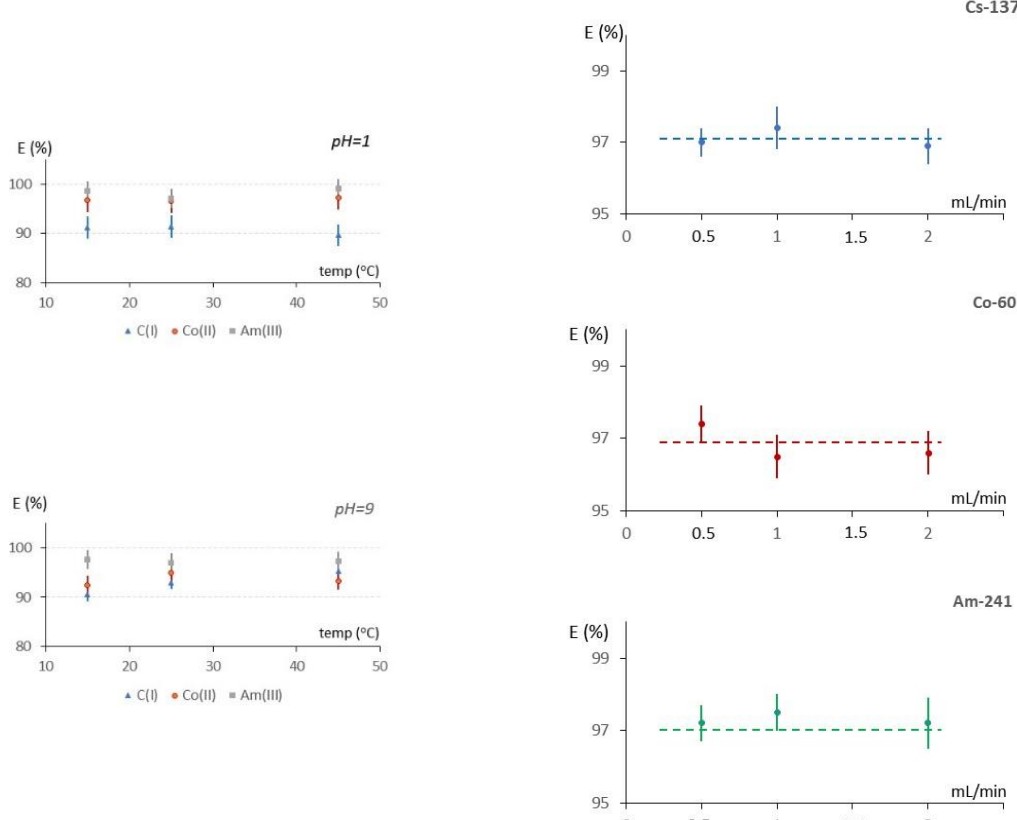

**Figure 5.** Temperature dependence of removal of Cs(I), Co(II) and Am(III) radionuclides from wastewater (**left**) and the purification efficiency, E (%), as a function of flow rate of the solution (**right**).

To answer the question of if the flow rate of water passing through the UF membrane affects the efficiency of purification, we compared the purification efficiency of water at pH = 5 for three different flow rates forced by the increased pressure of the nitrogen stream. The results shown in Figure 5 (right) demonstrate that the flow rate increase from 0.5 mL·min$^{-1}$ to 2 mL·min$^{-1}$ does not significantly affect the uptake of the radionuclides.

Membrane fouling is the process by which components of the solution or of the suspension deposit on the membrane surface or in the pores. Therefore, the filtration properties of the membrane decrease, and, finally, the process is stopped [26]. Hence, to check the separation capacity of the fouled membrane, we filtrated through the membrane 1 L of the solution containing the radioactively pure CSS suspension with a flow rate of 10 mL·min$^{-1}$ prior to a typical sorbent-assisted UF of the feed solution containing the radionuclides. It was detected that the deposition of a CSS layer formed from the suspension does not significantly influence E%. The respective data are shown in Figure 6.

However, the effect of fouling on the permeate flow was observed. Thus, in order to maintain a constant flowrate of water through the installation, it was necessary to apply a gradual increase in the pressure of the nitrogen stream forcing the flow.

As was expected, a decrease in the permeate flux during each of the filtration experiment also occurred. An example of the changes in the permeate flux is presented in Figure 7. Compared to the flow of pure water through a new membrane (0.0173 m$^3$·m$^{-2}$s$^{-1}$), the decrease was evident.

However, after approximately the first 10 min of the operation, the permeate flux began to be stable at 70% of the original permeate flux.

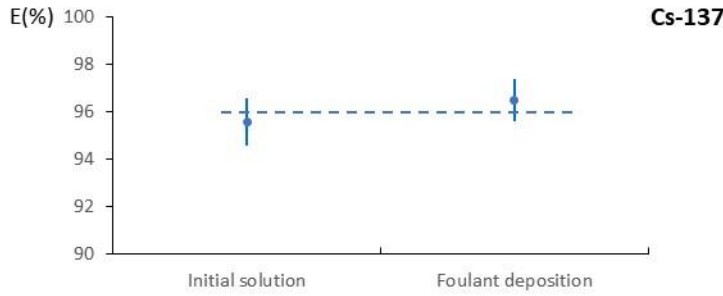

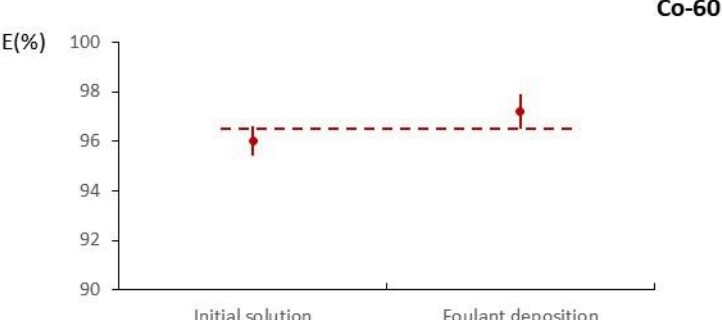

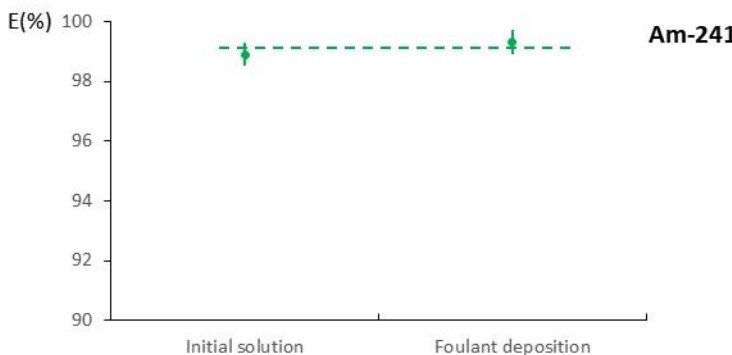

**Figure 6.** Changes in the purification efficiency, E (%), of three radioactive metal cations because of the membrane fouling.

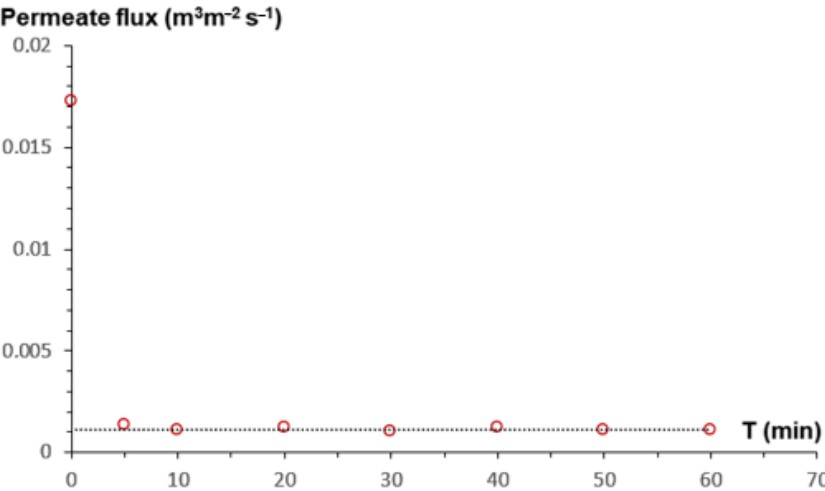

**Figure 7.** Changes in the permeate flux during the SAUF experiment.

3.1.2. Removal of Anionic Metal Species ($TcO_4^-$)

The radionuclide of the technetium-99 (Tc-99) is important from an environmental point of view because of its long half-life. It is usually produced in extremely small amounts as a result of the spontaneous fission of U-238, and it can be released into the environment in greater amounts by the nuclear fission of U-235, both as a result of nuclear reactor operation and military tests [27]. The amount of Tc-99 produced by the global nuclear industry until 1983 was estimated to be at around 15,000 TBq (24 tons) [28].

Technetium may exist in the solution in eight oxidation states: from −1 to +7. The latter is the most chemically stable and forms oxyanion of the pertechnetate ($TcO_4^-$), chemical analogue of its congener, a permanganate ion. However, most of the nuclear medicine procedures use it in a reduced form produced by the action of the stannous chloride ($SnCl_2$) [29]. Some inconvenience of using this reduced technetium form comes from its easy re-oxidizing in the presence of air and the re-forming of the pertechnetate species. The rate of this process increases with a growing air content in the system.

Therefore, we also tried to check the possibility of purifying contaminated water using the CSS-enhanced ultrafiltration method. The results of the removal of both the anionic- and the cationic forms of technetium-99m are presented in Figure 8. The volumes of the fractions marked in figure were 10 mL each.

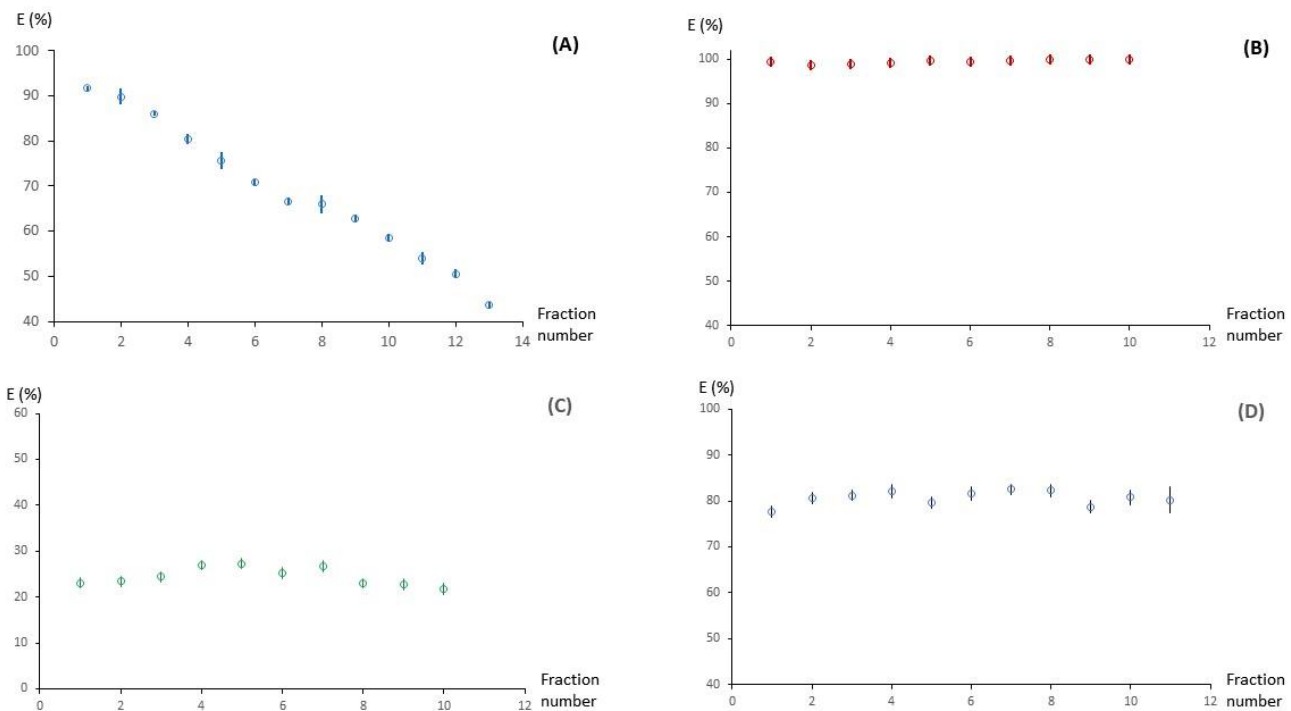

**Figure 8.** Removal of technetium-99m from wastewater: (**A**) natural form of Tc-99m, (**B**) $SnCl_2$ added, (**C**) water containing the complexing agents ($SnCl_2$ reduction), (**D**) hydrazine added.

As can be seen from Figure 8A, the removal of the pertechnetate in the non-reducing conditions and in the presence of air was unsatisfactory. As was already shown [30], oxyanions do not bind to the aluminosilicate and penetrate through the pores of the membrane as smaller species than the potentially technetium-saturated sorbent particles. Thus, even if immediately after contacting the solution to be purified with the CSS, some extent of the reduced technetium forms appears; soon after, they are re-oxidized, and the efficiency of the removal decreases. The addition of the reducing agent to the purified water seems to be the proper way to solve the problem.

Figure 8B shows, in turn, that, in the reducing conditions, the removal efficiency of technetium increases compared to pertechnetate. It seems to be predominantly due to the sorption of metal in the cationic form on the aluminosilicate sorbent. Thus, the addition of

the stannous chloride may be successively used for the decontamination of water containing technetium by the proposed SUF method. This observation is consistent with the results obtained earlier from the batch sorption of Tc-99m by the CSS as a single process [30]. Our previous studies have shown that the sorption of the pertechnetate anion by the CSS is negligible (i.e., the $E_{Tc}$ value is close to zero), but the addition of the $SnCl_2$ results in a significant increase in the technetium sorption ($E_{Tc}$ approaches 100%). However, if water contains the organic complexing agents (e.g., coming from the professional decontaminating solutions), the efficiency of the SAUF methods decreases dramatically (see Figure 8C). It can be suggested that this is a consequence of the pertechetate ions complexation, which stabilizes them and prevents their reduction.

As the supplementary experiments, we checked the usefulness of a "greener" reducing agent, such as hydrazine. Recently, Obruchnikova et al. have already shown the possibility of using hydrazine as the reducing agent for technetium(VII) [31]. As can be seen from Figure 8D, the presumption of using this agent as an alternative to $SnCl_2$ seems to be justified. The removal efficiency $E_{Tc}$ of the range of 80%, which is only slightly lower than for stannous chloride, suggests that using a green technetium reductor in the process of water purification from radioactive contamination is not only possible, but strongly recommended.

### 3.1.3. Treatment of the Simulated Liquid Radioactive Waste

As can be found in the Annual Report of the President of Polish Atomic Energy Agency, a volume of approximately 80% of the radioactive waste produced in 2021 was liquid waste [32]. It is known that liquid radioactive waste, not to mention industrial water, in addition to the radionuclides, contains dissolved salts. The content of these salts depends on the origin of the waste. The approximate average concentrations of the main salts in the Polish waste is: $CsNO_3$: 0.30 g·L$^{-1}$, $NaNO_3$: 1 g·L$^{-1}$, $KNO_3$: 0.6 g·L$^{-1}$, $MgCl_2$: 0.5 g·L$^{-1}$ and $CaCl_2$: 0.2 g·L$^{-1}$ [33].

Based on this information, we prepared a simulated waste solution (though free of the radionuclides) and compared the content of the salts in the permeate collected from our laboratory unit with that of the raw water. The quantity of the cations and anions was determined by the ion chromatography method, and the result is presented in Table 2:

**Table 2.** Chemical composition of simulated raw water and permeate from the CSS-supported UF.

| Cation */Anion | Raw Water | Permeate |
|:---:|:---:|:---:|
| | [mg L$^{-1}$] | |
| $Na^+$ | 1961.6 | 205.5 |
| $K^+$ | 752.6 | 73.1 |
| $Mg^{2+}$ | 121.1 | 5.8 |
| $Ca^{2+}$ | 18.1 | 1.3 |
| $Cl^-$ | 219.5 | 193.4 |
| $NO_3^-$ | 350.2 | 330.7 |

* Transition metals—under the limit of detection (LOD).

The data presented in Table 2 show that the CSS-assisted ultrafiltration method is effective in the removal of both mono- and divalent cations. Anions, in turn, are retained in the permeate, as was expected. The detailed values indicate that salt concentrations in the permeate meet the Polish requirements for its use as industrial and even drinking water [34]. The water also meets the requirements of the WHO, which are slightly less restrictive in this area [35].

### 3.2. Discussion of the Results in Relation to the Previous Outcomes

Sorption-assisted ultrafiltration (SAUF) is a hybrid process combining metal binding properties of the water-insoluble high-molecular species (sorbents) with the filtering ability of the membranes. The method was considered useful for the removal of the transition-

and heavy metal cations from the contaminated water and liquid industrial aqueous wastes [36,37]. The available literature on the possibility of the application of the method in radioactively contaminated water management until today is limited.

The hybrid SAUF method consists of two successive separation processes. In the first, adsorption, due to the binding of small-molecular metal ions with large-molecular sorbent species, conditions are formed for the separation of the solvent (water) and the metal ions contained in the feedstock. Ultrafiltration, in turn, is intended to separate the suspension into a pure solvent and a metal-containing concentrated phase. The purity of the resulting solvent is expected to be higher compared to conventional filtration. Thus, the selection of the efficient sorbent seems to be a crucial point for designing the SAUF process of the removal of the radionuclides from aqueous solutions. The presented work deals with testing the recently unknown sorbent, which can be used for the removal of radionuclides from water using the SAUF method.

Below, we discuss the results of our studies on using a purified aluminosilicate sorbent based on the clay–salt slimes (CSS), an industrial waste formed by the Joint Stock Company "Belaruskali". It was already successively tested as a low-cost, eco-friendly sorbent in the batch sorption and was found to be effective for purification aqueous solutions containing Cs(I), Sr(II), Eu(III) and Am(III) radionuclides [20,30,38]. The main physicochemical properties important for metal sorption, i.e., full mineralogical sorbent composition, scanning electron microscopy (SEM) images, $\zeta$-potential, the infrared vibrational spectra, X-ray diffraction pattern and micrographs of the clay particles, may be found in papers [20,38].

It has already been indicated that the CSS sorbent is mostly formed by illite (an alumina sheet sandwiched between two silica sheets. The individual structural units are bonded with one another by mainly non-exchangeable potassium ($K^+$) ions between them) (approximately 42% by weight), but it also contains dolomite (an anhydrous carbonate mineral composed of nominally equal parts of calcium and magnesium carbonates, $CaMg(CO_3)_2$) (25% by weight) and potassium microcline feldspars (the name denotes a number of minerals containing potassium: orthoclase (endmember formula $KAlSi_3O_8$), microcline (chemically the same as orthoclase, but with a different crystalline structure), sanidine ($(K,Na)(Si,Al)_4O_8$) and some others) (16.5% by weight) [38]. The plate-like structure of the main component, illite, is formed by the silica tetrahedron–alumina octahedron–silica tetrahedron 2:1 layers and has a large specific surface area. Thus, the high sorption capacity for cations is a result of both ion exchange and a surface complexation mechanism, which is shown in Figure 9.

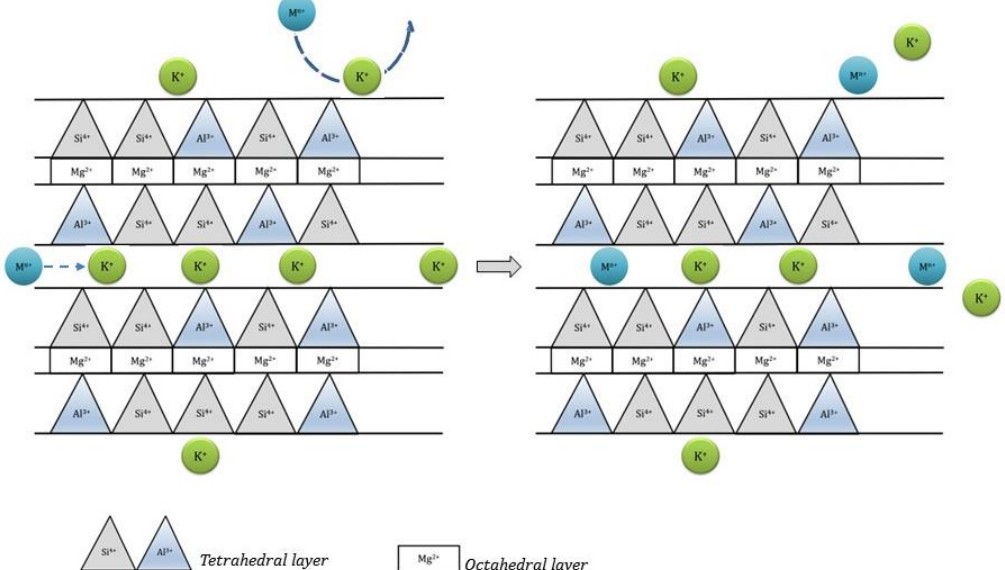

**Figure 9.** Schematic presentation of sorption mechanism of metals by illite.

To determine the optimum time of conducting the first process constituting the hybrid SAUF process (sorption), we recalled previously mentioned studies [30,38], and, more specifically, the parts concerning the effect of the sorbent contact time with the aqueous mixtures on the sorption efficiency. It has already been found that the metal removal efficiency increases rapidly in the first minutes and then quickly reaches a plateau. This fact may be related to changes in the concentration gradient between the sorbent and the liquid phase: at the beginning of the process, the large concentration gradient forces a large-value sorption rate. Then, the metal uptake reaches a plateau that corresponds to the insignificant gradient difference. Taking this into account in the presented work, we decided that 10 min is sufficient for performing the first step—sorption—in the SAUF experiments. It has also already been found that the pseudo-second-order model describes the best sorption of the radionuclides. Thus, the rate-determining step in the sorption stage is chemical sorption (chemisorption), which is caused by forming the valence forces between the sorbent and sorbate by sharing their electrons [39]. In turn, the speed of the water flow through the pores of the membrane, as a physical process, depends only on the applied pressure of the inert gas forcing this stage of the hybrid process.

The effect of the sorbent dosage on the SAUF removal of radioactive metal ions from water is one of the crucial parameters that should be considered when designing optimum process conditions. Our previous batch experiments on the effect of the sorbent dosage on the sorption efficiency have shown that the metal removal remains constant above the value of $2 \text{ g} \cdot \text{L}^{-1}$. As the efficiency of the metal ion removal may be related to the number of exchangeable sites of the sorbent, it may be assumed that the number of the metal-binding sites is much greater than the amount of metal species present in the solutions of the concentration below $10^{-6} \text{ M l}^{-1}$. The proposed optimum sorbent dosage of $2.5 \text{ g} \cdot \text{L}^{-1}$ in the present work remains consistent with the range established in the batch experiments and was used in further SAUF experiments (Figure 4).

The acidity of the purified water is a significant factor that determines the removal efficiency. This is due to the fact that the metal speciation and dissociation of the binding sites of the sorbent determine charges of the reacting species [40]. If we take into account the already obtained values of the CSS $\zeta$-potential, we may conclude that, both in acidic and in basic solutions, the values are slightly negative (do not differ significantly from $-25$ mV) and do not change in the course of the action of the ionizing radiation [30]. Such negative values demonstrate that CSS is a proper material for use as a sorbent of metal ions present in aqueous solutions.

Furthermore, results that show the pattern of the efficiency of the removal of tracers of the radioactive metal ions vs. pH studied by the batch sorption show that, within the pH range from 2 to 12, it does not change remarkably [20,30]. The same has been observed in the presented work for the SAUF experiments for either mono- or multivalent cations (Figure 3). It is known from the literature that the problem of the adsorption of heavy metals on the layered aluminosilicates, where CSS is an aluminosilicate, is caused by the metal interactions with their binding sites of different affinity towards these metals. Those of stronger affinity (ionic sites) are placed on the edge parts of the silicate layers and their amount increases with an increasing pH. The interactions of weaker binding sites (the silica- and alumina octahedra), in turn, are in the interlayer space [41]. Therefore, the observed poor dependence of the removal efficiency on the solution acidity could indicate that the CSS material adsorbs the metallic radionuclides primarily due to the radionuclide binding by the silica- and alumina octahedra.

The poor dependence of metal removal on the temperature and water flow rate (Figure 5) suggests that the ambient temperature of the process and the relatively slow penetration of water through the membrane pores should be maintained in the SAUF process because they do not require an additional energy input.

In this work, we also tested the possibility of the removal of technetium-90, one of the most dangerous for the environment and human health radionuclides, in the SAUF process that uses the CSS sorbent. As is well known, this radionuclide occurs in an anionic form,

e.g., as in the batch-sorption, removal of the anionic species in the SAUF method is close to zero (Figure 8). The addition of the reducing agents, however, results in a significant increase in technetium removal, as a result of its reduction to the cationic forms [42]. Finally, it approaches 100% (Figure 8). Such green reagents, as hydrazine or ascorbic acid, may be suggested for the SAUF procedure.

The same can be said about the radionuclides of manganese and chromium (i.e., Mn-54, Cr-51), which are products of the corrosion of the steel parts in the nuclear installations [43].

## 4. Conclusions

The paper presents the potential use of the CSS-based SAUF method to remove radionuclides in various forms from radioactively contaminated aqueous solutions. The work makes it evident that the tested configuration of SAUF may be used for the decontamination of the radioactively contaminated water, as well as for the treatment of liquid radioactive waste. The use of waste material from the fertilizer production, namely CSS, as a radionuclide sorbent creates prospects for its effective utilization.

**Author Contributions:** Conceptualization, L.F., A.M. and G.Z.-K.; methodology, L.F., A.M. and G.Z.-K.; investigation, L.F. and A.M.; writing-original draft preparation, L.F. and A.M.; writing-review and editing, L.F., A.M. and G.Z.-K.; supervision, G.Z.-K. All authors have read and agreed to the published version of the manuscript.

**Funding:** This research was funded by the IAEA Research Contract No. 23022/R0 in the frame of the Coordinated Research Project Management of Wastes Containing Long-lived Alpha Emitters: Characterization, Processing and Storage.

**Institutional Review Board Statement:** Not applicable.

**Informed Consent Statement:** Not applicable.

**Data Availability Statement:** Not applicable.

**Acknowledgments:** The authors are grateful to Leanid Maskalchuk (Belarusian State Technological University Minsk, Belarus) for providing us the CSS sorbent and acknowledge Miroslaw Buta (Institute of Nuclear Chemistry and Technology) for help in the SAUF experiments.

**Conflicts of Interest:** The authors declare that there is no conflict of interest in the presented work.

**Sample Availability:** Samples of the compounds are available from Leanid Maskalchuk (Belarusian State Technological University Minsk, Belarus).

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
