# Peer review of "Sorption-Assisted Ultrafiltration Hybrid Method for Treatment of the Radioactive Aqueous Solutions"

_chemistry, doi:10.3390/chemistry4030073_

Round 1
Reviewer 1 Report
In the present work, the authors introduced a method of decontamination of radioactive waste containing water by a sorption-assisted UF in which the clay-salt slimes (ccs) were utilized as a sorbent in the range from 0.5 to 3.0 191 gˑL-1. Overall, the manuscript is presented nicely but needs to improve before getting published. See below for the critical comments to improve the manuscript.
· The authors have to cite some references in this claim and also the motif is missing in this paragraph (Page 2). "The possibility of separating small ions by binding them to water-soluble polymers (e.g. chitosan, sodium alginate) has been proposed and studied".
· Is Figure 1 created by the Authors? If not, please cite the actual work.
· The authors used polyether sulfone membrane for the ultrafiltration process, do the authors measure the actual flux of the water with and without the presents of CCS?
· Please also emphasize the fouling of the membrane on a qualitative method through measuring actual water flux experiments. The practical applicability of these kinds of sorbents is largely limited by this phenomenon. Therefore, the authors need to explain this matter.
Author Response
1.The authors have to cite some references in this claim and also the motif is missing in this paragraph (Page 2). "The possibility of separating small ions by binding them to water-soluble polymers (e.g. chitosan, sodium alginate) has been proposed and studied".
Respective references have been included.
2. Is Figure 1 created by the Authors? If not, please cite the actual work.
Figure 1 was prepared by the authors of this paper and was based on an extract of the information that can be found in the literature of the problem.
3. The authors used polyether sulfone membrane for the ultrafiltration process, do the authors measure the actual flux of the water with and without the presents of CCS?
The water flux (Jw) was determined experimentally prior to the UF/sorption experiment under transmembrane pressure of 2 bar. The measured Jw value for clean membrane and distilled water was 0,0173 m3 m-2 s-1). This missing information has been added in the manuscript.
4. Please also emphasize the fouling of the membrane on a qualitative method through measuring actual water flux experiments. The practical applicability of these kinds of sorbents is largely limited by this phenomenon. Therefore, the authors need to explain this matter.
The membrane fouling phenomenon has been already discussed in the manuscript (pages 9-10).
It was emphasized that there was an effect of fouling on the permeate flow , which implied the need to increase the pressure in order to maintain a constant flow-rate of water. Obviously, the study on the fouling phenomenon by measuring the actual permeate flux was also performed. Example of the results of this study has been added to the manuscript in the form of. Fig.7. The figure shows also the value of the permeate flux determined for a clean membrane and distilled water, which marked the beginning of the experiment.
Reviewer 2 Report
The current manuscript describes the “Sorption-assisted ultrafiltration hybrid method for treatment of the radioactive aqueous solutions”. In this study, the work represents the results of the studies on the possibility of using the ultrafiltration method supported with sorption on the low-cost, easily accessible aluminosilicates to purify water contaminated with radionuclides or the low-level radioactive waste treatment. The UF membrane's performance was studied as obtained results show that hazardous impurities, both in the form of cations and anions, may be effectively removed from water by application of sorption-assisted UF using the clay-salt slimes as a sorbent. After reviewing the manuscript reviewer pointed out that there is a lot of scope for improvement of this manuscript. A lot of sections may need further illustration; and more importantly, because this type of work has been already reported by a lot of researchers using different types of sorption materials. One important point is that the abstract need to be revised completely.
The authors should consider critically these comments to improve the quality of the work.
Few examples of where authors should make changes are provided below:
- The abstract is not showing the exact findings and output and what the authors want to show here in short. Authors have to rewrite the abstract.
- No mechanism has been explained in the paper properly with citations. Authors have to explain it properly.
- Line 56 has no reference for this sentence as evidence for water-soluble polymers (e.g. chitosan, sodium alginate).
- The introduction has to scope to revise there is no need for tables and figures in the manuscript. Authors can add it in the discussion part. In the introduction after reference 16, this sentence the reference needs to be there as per discussion like International Journal of Biological Macromolecules Volume 193, 2021, Pages 2121-2139; Chemical Engineering Journal Volume 446, 2022, 137303.
- Try to add a few characterizations of the materials.
- Figure resolutions have to be checked and add high-resolution figures.
- This is the chemistry journal that needs to add some chemistry of materials.
- All equations need to check and need the equation number in the manuscript.
- Authors have to look always starting words of the sentences.
- Grammar needs to check one more time.
Author Response
1. The abstract is not showing the exact findings and output and what the authors want to show here in short. Authors have to rewrite the abstract.
Text of the abstract has been changed according to the reviewer's note.
2. No mechanism has been explained in the paper properly with citations. Authors have to explain it properly.
The new MS version includes a revised section entitled Discussion of the results in relation to the previous outcomes. This version takes into account the suggestions proposed by the reviewers.
3. Line 56 has no reference for this sentence as evidence for water-soluble polymers (e.g. chitosan, sodium alginate).
Respective references have been included.
4. The introduction has to scope to revise there is no need for tables and figures in the manuscript. Authors can add it in the discussion part. In the introduction after reference 16, this sentence the reference needs to be there as per discussion like International Journal of Biological Macromolecules Volume 193, 2021, Pages 2121-2139; Chemical Engineering Journal Volume 446, 2022, 137303.
The text of the introduction has been changed so, that it meets the requirements of the Reviewer.
Data from the Table 1 have been inserted to the text and Figure 1 has been transferred to the discussion part of the paper.
5. Try to add a few characterizations of the materials
The physicochemical properties and the mineralogical composition of CSS have been already described in the published papers, so the relevant references have been given.
6. Figure resolutions have to be checked and add high-resolution figures.
We will attach the JPG files of the required resolution.
7. This is the chemistry journal that needs to add some chemistry of materials.
Discussion part has been changed and enlarged.
8. All equations need to check and need the equation number in the manuscript.
In the whole manuscript we use only one equation, so we did not assign a number to it. However, because of the request of the reviewer, we have inserted this number. The equation is correct.
9. Authors have to look always starting words of the sentences.
The text has been checked for the grammar.
10.Grammar needs to check one more time.
The text has been checked for the grammar.
Round 2
Reviewer 1 Report
The authors' responses to the comments are satisfactory and the manuscript has been improved with many corrections and additions. Considering these, I believe the revised manuscript is publishable in its current form!
Reviewer 2 Report
Accept with minor revision as commented.
Authors have not resolved the below 3 comments from the previous version.
- Line 56 has no reference for this sentence as evidence for water-soluble polymers (e.g. chitosan, sodium alginate).
- The introduction has to scope to revise there is no need for tables and figures in the manuscript. Authors can add it in the discussion part. In the introduction after reference 16, this sentence the reference needs to be there as per discussion like International Journal of Biological Macromolecules Volume 193, 2021, Pages 2121-2139; Chemical Engineering Journal Volume 446, 2022, 137303; Journal of Membrane Science Volume 609, 2020, 118212; Membranes 12 (8), 2022, 768 etc.
- Try to add a few characterizations of the materials.